# Position: General Intelligence Requires Reward-based Pretraining

Seungwook Han [* 1]  Jyothish Pari [* 1]  Samuel J. Gershman [2]  Pulkit Agrawal [1]

## Abstract

Large Language Models (LLMs) have demonstrated impressive real-world utility, exemplifying artificial useful intelligence (AUI). However, their ability to reason adaptively and robustly – the hallmarks of artificial general intelligence (AGI) – remains fragile. While LLMs seemingly succeed in commonsense reasoning, programming, and mathematics, they struggle to generalize algorithmic understanding across novel contexts. Our experiments with algorithmic tasks in esoteric programming languages reveal that LLM's reasoning overfits to the training data and is limited in its transferability. We hypothesize that the core issue underlying such limited transferability is the coupling of reasoning and knowledge in LLMs.

To transition from AUI to AGI, we propose disentangling knowledge and reasoning through three key directions: (1) pretaining to reason using RL from scratch as an alternative to the widely used next-token prediction pretraining, (2) using a curriculum of synthetic tasks to ease the learning of a *reasoning prior* for RL that can then be transferred to natural language tasks, and (3) learning more generalizable reasoning functions using a small context window to reduce exploiting spurious correlations between tokens. Such a reasoning system coupled with a trained retrieval system and a large external memory bank as a knowledge store can overcome several limitations of existing architectures at learning to reason in novel scenarios.

## 1. Introduction

Large Language Models (LLMs) have demonstrated impressive capabilities across diverse tasks, such as commonsense

[*]Equal contribution  [1]Improbable AI Lab, MIT [2]Department of Psychology and Center for Brain Science, Harvard University. Correspondence to: Seungwook Han <swhan@mit.edu>, Jyothish Pari <jyopari@mit.edu>.

*Proceedings of the 42$^{nd}$ International Conference on Machine Learning*, Vancouver, Canada. PMLR 267, 2025. Copyright 2025 by the author(s).

reasoning, math, and programming (Ho et al., 2022; Wei et al., 2023; 2022; Lampinen et al., 2022; DeepSeek-AI, 2024; OpenAI, 2024; Meta, 2024). Their practical impact makes them a compelling instance of *artificial useful intelligence* (AUI) – systems that assist humans in real-world tasks. Nevertheless, a significant gap remains between AUI and artificial general intelligence (AGI), systems capable of robust, adaptive reasoning across diverse domains and contexts like humans.

To illustrate this gap, we designed algorithmic tasks in esoteric programming languages that isolate reasoning from memorization. These tasks involve simple algorithmic problems (e.g., printing, sorting) seen during pretraining and easily solved in Python and Java, but presented in unfamiliar programming languages with different syntaxes. Our results in Section 3 show that state-of-the-art LLMs struggle to transfer their algorithmic understanding to coding in new programming syntaxes. Notably, o1 (Jaech et al., 2024), post-trained for reasoning with a combination reinforcement learning (RL) and search, e.g., Chain-of-Thought (Wei et al., 2023), performed the best. However, even o1's performance suffers, illustrating the limitation of current models in flexibly transferring their reasoning to novel contexts.

We hypothesize that going beyond the reasoning ability of current models requires a fundamental overhaul of the pretraining paradigm. The dominant approach–supervised pretraining with next-token prediction loss on passively collected Internet data, followed by RL-based post-training (DeepSeek-AI, 2025; Lightman et al., 2023; Luong et al., 2024; Zelikman et al., 2024; 2022)–closely mirrors AlphaGo's methodology in the game of Go (Silver et al., 2016). In AlphaGo, supervised pretraining on human demonstrations and post-RL optimization were used to surpass human playing abilities. However, this paradigm was overturned by AlphaZero (Silver et al., 2017), which demonstrated that RL from scratch– without supervised pretraining– achieved superior performance. By relying solely on self-play and starting from random initialization, AlphaZero surpassed AlphaGo, uncovering more efficient and creative strategies through purely RL.

This paradigm shift from AlphaGo to AlphaZero motivates our hypothesis that, even in current LLMs, reliance on supervised pretraining constrains models to a "local minimum" of reasoning capabilities. Supervised pretraining on demon-

stration data can be a double-edged sword: while it provides a helpful exploration bias for reasoning finetuning via RL, it also serves as a bias that may hinder subsequent exploration. Consequently, we hypothesize that next-token prediction training on Internet-scale data may constrain models to a local minimum of reasoning abilities that cannot easily be escaped through post-training RL.

Our intuition behind why next-token prediction might lead to a local minimum in reasoning is as follows: when the training objective is merely to predict the next token, the model can exploit correlations between tokens in the context window to reduce prediction error instead of learning the underlying reasoning algorithm that explains the data. For instance, if the model is trained on examples such as $5+3 = 8; 2 + 1 = 3; 4 + 7 = 11$, it might learn that the next character after the character $+$ and a *number* is the character $=$. In other words, the model can learn superficial statistical patterns (e.g., which words commonly occur together) rather than developing genuine understanding (e.g., addition) that will generalize across different situations.

As another example, consider instructing a model to write Python code with 1-based indexing instead of the usual 0-based indexing. To solve the task with the simple change, the model must override its memorized Python knowledge and flexibly apply the 1-based indexing rules learned from other languages like R or MATLAB. However, many models pretrained with next-token prediction produce 0-indexing based solutions, leaning on surface-level pattern matching that ignores the instruction (Wu et al., 2024). This illustrates how passive pretraining fails to incentivize the learning of generalizable reasoning skills and, instead, reinforces memorization and pattern matching that have limited generalization beyond the training scenarios. When a model learns to reason in a way that is *tied to specific ways of expressing things rather than underlying principles*–the model struggles to apply its reasoning to new scenarios.

**In this position paper, we argue that to advance from AUI to AGI, it is critical to deliberately disentangle knowledge from reasoning, allowing models to develop robust reasoning independent of memorized patterns. Additionally, we propose architectural modifications that enable the reasoning system to flexibly adapt to newly added knowledge, ensuring that models can generalize their reasoning strategies to novel domains. To this end, we outline three key directions to achieve this paradigm shift, which we discuss in detail in Section 4.**

1. **Pretraining for reasoning via reward-based learning (Section 4.1):** The choice of training data can significantly influence whether the model exploits correlations to learn fragile reasoning or discovers the more general underlying reasoning algorithm. As an illustration, consider training a model to multiply numbers with data of the form $[2 \times 5 = 10; 3 \times 4 = 12; 5 \times 7 = 35]$, where the answers are directly provided. We refer to such data as *passive data* that is the outcome of human knowledge and reasoning (i.e., most of the Internet data) but doesn't encompass the reasoning process. Continuing with the analogy of AlphaGo, human gameplay data used for supervised training is *passive* in the sense that it contains *what* moves humans played but *not why* those moves were played. While one hopes that a model can implicitly learn the true reasoning process if trained with the next-token prediction objective, the model could memorize the answers or exploit correlations in the data to learn reasoning that works for in-distribution data, but fails beyond the training data distribution.

   Consider the alternative of training model on data that spells out the reasoning algorithm (i.e., *reasoning trace*) – $[2 \times 5 = 5 + 5 = 10; 3 \times 4 = 4 + (4 + 4) = 12; 5 \times 7 = (7+7)+(7+7)+7 = 35]$. We hypothesize that the model has a better chance of learning the underlying algorithm with such data. However, obtaining large amounts of *reasoning trace* data is challenging. One way is to use RL to generate data that maximizes the task reward (i.e., correct multiplication or winning the game of Go). In other words, by structuring training as an iterative (Carreira et al., 2016) and step-wise problem-solving process rather than next-token prediction on passive data, models can develop more robust and generalizable reasoning abilities not limited to the spurious patterns observed in pretraining data.

2. **Enabling Efficient Exploration via Synthetic Tasks (Section 4.2):** Building on the insight that data containing reasoning traces may better enable the model to learn the underlying reasoning process compared to *passive* data, the natural question is how to obtain such data. We propose that this data can be obtained via a more active process where an agent optimizes task rewards through a sequence of actions and generates reasoning traces in the process. However, as natural language comprises of ∼40K tokens, doing RL from scratch is infeasible. To address the exploration challenge, we propose using synthetic tasks with a reduced token space while preserving key structural properties to learn a *reasoning prior* (e.g., commutativity, associativity, in-context learning, etc.). The token space and complexity can gradually be increased via a task curriculum that increases difficulty. Once the reasoning prior is learned, it can leveraged to bootstrap reasoning in natural language.

3. **Architectural bias to decouple knowledge and reasoning (Section 4.3):** Training models using long context windows provide more chances for the model to learn *spurious correlations* between past and future tokens when minimizing the next-token prediction loss.

Proposed Training Paradigm

Proposed Architecture

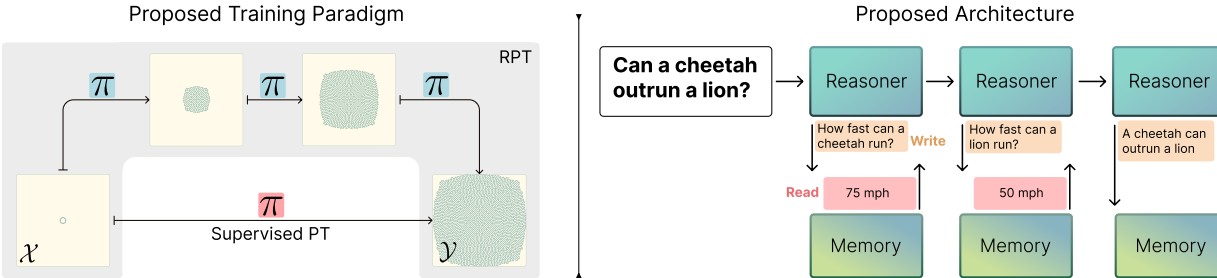

*Figure 1.* (Left) Comparing reward-based pretraining (RPT) and traditional supervised pertaining (SPT). Supervised pretraining requires the model to directly predict answers, limiting its ability to refine intermediate solutions. In contrast, RPT enables step-by-step reasoning from the outset, allowing the model to iteratively approximate complex functions through simpler transitions that are easier to learn and more robust to errors (Carreira et al., 2016). (Right) Illustration of our proposed decoupled memory-reasoning architecture. This design allows the reasoning module to operate on shorter context windows that reduce the chances of learning spurious correlations and, thereby, more transferrable reasoning. The use of a shorter context window also encourages the model to learn how to dynamically read and write to memory, which facilitates the use of reasoning model on new problems and knowledge domains.

Previous work highlighted issues such as *lost-in-the-middle* phenomenon (Liu et al., 2024), where models become overly sensitive to token positions within a long context rather than learning a robust, transferable reasoning process. Drawing on insights from cognitive science–which emphasize the efficacy of a limited "working memory" (Miller, 1956; Elman, 1993)–we hypothesize that restricting the model to reason over only a *limited set* of tokens reduces the chances of exploiting spurious correlations between tokens and thereby promote more robust reasoning that can transfer to new knowledge domains.

To this end, we propose three architectural changes. We first disentangle knowledge and reasoning into distinct modules – an external memory bank and a reasoning network. Such decomposition provides an inductive bias for re-using the reasoning model on new knowledge domains. Second, we propose that the reasoning model should operate over a small context window, an inductive bias that reduces the chances of relying on spurious correlations for making accurate predictions. Thirdly, because the reasoning model operates on a short context window, it needs to retrieve and write information from the external memory bank. We propose to learn the strategies for reading from and writing to an external memory bank. We hypothesize that this structured approach enables dynamic retrieval and reasoning, reduces reliance on spurious correlations, and therefore improves generalization.

## 2. Background and Notation

Current LLMs are pretrained for next-token prediction over a large text corpus. Let $\mathcal{D}$ denote a large unlabeled text corpus collected from the Internet, which we refer to as **passive data**. This passive data includes books, articles, and online content and generally does not describe interme-diate reasoning steps that can guide problem-solving. For example, while a text passage might contain the statement of a math problem and its final answer, it often omits the step-by-step derivation leading to the solution (e.g., interme-diate algebraic manipulations or deductions). Each element in $\mathcal{D}$ is a sequence of tokens $\mathbf{x} = (x_1, x_2, \ldots, x_T)$, typi-cally formed using a vocabulary of size $|V|$. A language model $\phi_\theta$ with parameters $\theta$ (often a Transformer (Vaswani et al., 2023)) is trained to learn the conditional distribution $P(x_t \mid x_1, x_2, \ldots, x_{t-1}; \theta)$.

In practice, we maximize the log-likelihood of each token given its preceding tokens:$\max_\theta \sum_{\mathbf{x} \in \mathcal{D}} \sum_{t=1}^{T} \log P(x_t \mid x_{<t}; \theta)$. We refer to this task-agnostic pretraining using next-token prediction loss as **supervised pretraining** (SPT). Any subsequent training that continues to optimize this objective for a target task is referred to as **supervised fine-tuning** (SFT).

Training for tasks can also be performed using a reward-based formulation grounded in RL. At each step $t$, the model $\phi_\theta$ receives a scalar reward $r_t$, which can depend on partial or complete outputs. Here, the data is not the passively collected corpus $\mathcal{D}$, but rather is gathered **online** from the model's own interactions. The goal is to maximize the expected cumulative reward: $\max_\theta \mathbb{E}_{\mathbf{x} \sim \pi_\theta}\left[\sum_{t=1}^{T} r_t(\mathbf{x})\right]$, where $\pi_\theta$ is the model's policy distribution over sequences $\mathbf{x}$. We refer to the pretraining done with this reward-based objective as **reward-based pretraining** (RPT), and sub-sequent training that continues to optimize this reward as **reward-based finetuning** (RFT).

## 3. Evaluation of Reasoning Separate From Knowledge

To evaluate the ability of current models to transfer their rea-soning across domains, we constructed an evaluation bench-mark containing simple algorithmic tasks, such as printing

| Models | Print (# ICL = 1) | Print (4) | Print (10) | Sort | Copy |
|---|---|---|---|---|---|
| Llama 3.1 8B | 0.01 | 0.04 | 0.07 | 0.00 | 0.00 |
| Llama 3.1 70B | 0.02 | 0.07 | 0.10 | 0.00 | 0.00 |
| Qwen2.5 Coder 7B | 0.02 | 0.02 | 0.01 | 0.00 | 0.00 |
| Qwen2.5 Coder 32B | 0.03 | 0.02 | 0.02 | 0.00 | 0.09 |
| GPT-4o | 0.02 | 0.09 | 0.13 | 0.00 | 0.00 |
| o1 | 0.71 | 0.64 | 0.65 | 0.01 | 0.95 |

*Table 1.* Brainf\*\*k evaluation results. We report accuracy (in decimal format) over the test set for each task. The number of examples used for each task evaluation is listed in Appendix A.3. The numbers within parentheses are the number of in-context examples during evaluation.

and sorting, in esoteric programming languages. Unlike conventional evaluation using programming languages with familiar syntax, such as Python or Java, this benchmark is designed to minimize the influence of pre-existing knowledge by using languages with unconventional rules and minimal semantic connections to common programming languages. This allows us to measure a model's ability to generalize its logical reasoning to solve problems in domains different from the ones encountered during pretraining.

### 3.1. Esoteric Programming Languages

We used two esoteric programming languages, Brainf\*\*k (Wikipedia, 2025) and Befunge (Cat's Eye Technologies, 1993). Both languages are infrequently seen during pretraining and radically different from common languages like Python. However, they are Turing-complete, meaning they can express any algorithm that a programming language like Python can, given enough time and memory. Brainf\*\*k operates on a simple, tape-based memory model with only eight commands. Befunge is a two-dimensional stack-based language where code execution follows paths on a grid, allowing for new control flows like loops and branches in any direction. While both languages differ from Python, they are based on simple programming rules. Example below.

```
+++++++[>++++++++++<-]>+++++.
```

*Listing 1.* Brainf\*\*k program to print the character 'K.'

**Tasks.** We designed two sets of tasks for Brainf\*\*k and Befunge, which are easy to do in Python and Java. We select the tasks based on what is more suitable to implement in each esoteric language while maintaining some overlap. For Brainf\*\*k, we have the three tasks of printing two-letter words, sorting a list of five elements, and copying an input string of five characters. For Befunge, we have the three tasks of printing two-letter words, generating a program to calculate the factorial of the input number, and generating a program to output the first $k$ Fibonacci numbers.

**Evaluation Details.** We evaluate the state-of-the-art LLMs of different families and scales: Llama 3.1 8B and

70B, Qwen2.5 Coder 7B and 32B, GPT-4o and o1. For each task, we construct a set of problems by changing the inputs. For example, in the printing task, we vary the characters to print (e.g., 'hi' and 'so'). The models are prompted using the standardized format reported in Listing 5 and Listing 6 of Appendix A.4 and A.5 respectively.The prompt includes the full syntax and rules of the programming language and example code blocks with explanations. We further detail the evaluation protocol in Appendix A.3.

**Results.** Despite the simplicity of the tasks, all models generally perform poorly, averaging ∼12% accuracy in Brainf\*\*k (see Table 1) and ∼29% in Befunge (see Table 2). In Brainf\*\*k, performance only marginally improves (about 4% on average) with the increase in the number of in-context examples from 1 to 10 suggesting that current models struggle to infer the correct structure and principles underlying Brainf\*\*k, even with all the rules, syntax, and contextual guidance. On the other hand, in Befunge, in-context examples allow the models to achieve ∼70-90% accuracy at printing two-letter words. But, the models fail to solve the Fibonacci and factorial tasks.

A notable outlier is the o1 model, which vastly outperforms other models. Unlike the other models, o1 has undergone extensive post-training with RL for solving reasoning tasks (Jaech et al., 2024), which likely contributes to its stronger performance. This result highlights the potential benefits of RL-based post-training in adapting models to unconventional tasks. However, we note that even o1's performance leaves considerable room for improvement. It scores 1% accuracy on sorting five elements even with 10 in-context examples and 65.5% accuracy on printing two-letter words.

## 4. Proposed Directions

### 4.1. Pretraining for Reasoning with RL

> **Proposal**: Instead of pretraining on passive data (defined in Section 2) and finetuning with RL, we propose integrating RL directly into the pretraining phase to enable better iterative reasoning.

| Models | Print (# ICL = 1) | Print (4) | Print (10) | Fibonacci | Factorial |
|---|---|---|---|---|---|
| **Llama 3.1 8B** | 0.19 | 0.68 | 0.72 | 0.00 | 0.00 |
| **Llama 3.1 70B** | 0.92 | 0.92 | 0.93 | 0.02 | 0.00 |
| **Qwen2.5 Coder 7B** | 0.15 | 0.66 | 0.74 | 0.00 | 0.00 |
| **Qwen2.5 Coder 32B** | 0.08 | 0.83 | 0.94 | 0.00 | 0.00 |
| **GPT-4o** | 0.72 | 0.98 | 1.00 | 0.00 | 0.00 |
| **o1** | 0.70 | 0.83 | 0.93 | 0.00 | 0.00 |

*Table 2.* Befunge evaluation results. We report accuracy (in decimal format) over the test set for each task. The number of examples used for each task evaluation is listed in Appendix A.3. The numbers within parentheses are the number of in-context examples during evaluation.

Recent works have continued the SPT-then-RFT paradigm to enhance task adaptability and reasoning capabilities in LLMs. In particular, RFT has shown to be essential in improving generalization across reasoning-intensive domains, such as mathematical problem-solving and program synthesis (Zelikman et al., 2024; Hosseini et al., 2024). By enabling and refining intermediate reasoning traces (e.g., "Let's solve this step by step: First, we need to factor the quadratic equation...") to arrive at a final answer, these methods decompose complex problems into easier sub-problems, allowing models to iteratively construct solutions through systematic exploration of possible solution paths.

The current approach of training LLMs with the SPT-then-RFT paradigm mirrors AlphaGo (Silver et al., 2016), which initially leveraged pretraining on human demonstrations, followed by RL finetuning, achieving superhuman performance in Go. However, AlphaZero (Silver et al., 2017), which trained purely with RL from scratch, surpassed AlphaGo, suggesting that imitation-based pretraining may limit exploration. A similar risk arises in LLMs: an initial phase of supervised pretraining on passive data (as defined in Section 2) often lacks the supervision of intermediate reasoning steps and may confine subsequent RL finetuning to a restricted solution space, hindering its ability to escape the local minimum. Moreover, it is common to constrain RL finetuning to stay close to the pretrained model to reduce the risk of generating unnatural sequence of language tokens and thereby hacking the reward function (Gao et al., 2023; Paulus, 2017; Alami et al., 2024). However, such a constraint also hinders the exploration of RL finetuning and its ability to discover the underlying reasoning process.

> **Hypothesis**: Pretraining on passive data can constrain the subsequent finetuning by placing models in local minima, limiting their ability to discover reasoning strategies that generalize (e.g., finding novel ways to solve math problems beyond the specific solution approaches seen in training data).

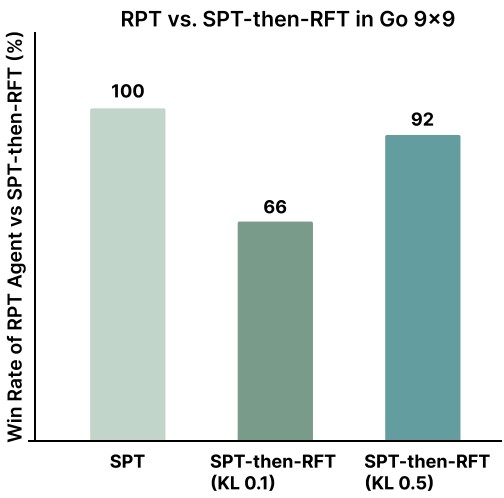

*Figure 2.* Comparing the different training paradigms of RPT vs. SPT-then-RFT in Go $9 \times 9$. These results affirm the hypothesis that the SPT can restrict the model's subsequent exploration with RL. The KL regularization is added to replicate common training paradigms (Gao et al., 2023; Paulus, 2017; Alami et al., 2024).

**Does reward-based pretraining (RPT) outperform supervised pretraining followed RL finetuning (SPT-then-RFT) in Go?**

We trained two agents using the **SPT-then-RFT** paradigm (as used in AlphaGo) and the **RPT** paradigm (as used in AlphaZero) in a simplified Go environment using a 9×9 board. See training details in Appendix B.1. This setup allows rigorous evaluation of whether pretraining on expert human demonstrations collected from the top 100 players impairs RL's ability to escape the local minima found by supervised learning and thereby restricts its exploration and ability to learn superior gameplay strategies. The AlphaZero paper introduced multiple changes from AlphaGo (e.g., different network architecture, pretraining strategies, etc.), making it challenging to pinpoint the most critical change responsible for the performance difference. We ran controlled experiments to isolate whether the RL training from scratch was the major factor contributing to performance gain. To

mimic the common training setup in LLMs, we adopt a Kullback-Leibler (KL) penalty (Kullback & Leibler, 1951) as a regularization for the SPT-then-RFT paradigm.

Go $9 \times 9$ results, reported in Figure 2, show that supervised pretraining on passive data constrains the RL finetuning performed similarly to RLHF in LLMs by limiting exploration. As expected, the RPT model achieves a 100% win rate against the SPT, demonstrating that exploration via reward-based training can easily outperform training on the top 100 expert data. Against the SPT-then-RFT paradigm, the performance of the RPT model varies with the KL constraint that keeps the generations close to the pre-trained model. The RPT model achieves a 66% win rate compared to SPT + RFT with a KL coefficient of 0.1, and a greater win rate of 92% against SPT + RFT with a stricter KL coefficient of 0.5. These findings highlight that tighter reliance on pre-trained knowledge hinders exploration while loosening these constraints enables RL to discover better strategies.

Without the KL constraint and given an infinite training budget, RPT and SPT + RFT models can eventually converge to the same optimal policy. Such recovery in performance using RFT after SPT may be possible in simpler tasks such as Go9x9 where reward hacking is not possible. However, running unconstrained RL post-SPT with LLMs is generally infeasible as the LLM degrades and starts producing gibberish (i.e., meaningless sequence of tokens) that still increases the reward function (i.e., reward hacking (Skalse et al., 2022)). To verify if this is ndeed the case we repeat the comparison between SPT and SPT + RFT with LLMs as detailed below.

**Does reward-based finetuning (RFT) outperform supervised pretraining followed by reward-based finetuning (SPT-then-RFT)?**

We test the hypothesis that, in LLMs, RL training for reasoning (RFT) after supervised pretaining may be insufficient to push the model beyond its local minimum to discover more general reasoning abilities that could have been learned if the model was trained from scratch to reason via RL.

As the training of a language model from scratch using RL remains very challenging (see Section 4.2 for more discussion), we use a pretrained LLM and test if supervised fine-tuning (SFT) on a passive dataset (i.e., examples of questions and answers) limits generalization compared to finetuning for reasoning using RL (RFT).

To make a fair comparison, we fix all variables constant except the training paradigms of: (1) **SFT-then-RFT** paradigm, where a base model is finetuned on supervised demonstrations before applying RL (Ziegler et al., 2019); (2) **pure RFT** paradigm, where the base model is only finetuned with a reward-based objective via RL. While this setup does not involve full SPT vs. RPT comparisons, it provides a computationally efficient proxy for understanding how SFT on passively collected data influences the exploration and generalization of downstream RFT.

For this investigation, we design a synthetic mathematical reasoning task wherein the model must identify vectors orthogonal to a given vector (more details in Appendix A.6). As shown in Listing 2, the problem is presented as a multiple-choice question, where the model must choose the correct option. We used Qwen 1.5B (Yang et al., 2024) as the base pre-trained model and evaluated different fine-tuning strategies using the prompt detailed in Appendix A.7.

*Listing 2.* Example of synthetic orthogonality task

```
Question: Which of the vectors are orthogon
-al to [-2, -1, 0]?
(a) [3, 4, -5]
(b) [0, 0, 2]
(c) [-1, 3, 4]
(d) [-1, 1, 5]

Answer: (b)
```

Results in Table 3 support our hypothesis: models trained with only RFT outperform those finetuned with SFT-then-RFT. The SFT-then-RFT models overfit to the training distribution, achieving near-perfect accuracy on the training set but inferior generalization to the test set reflected in a performance drop from 100% to 80%. This result suggests that the model is more prone to memorizing patterns in the training data instead of figuring out the underlying reasoning algorithm. These results are aligned with those of Chu et al. (2025) that show SFT memorizes while RL generalizes. Furthermore, SFT-trained models struggle to leverage RL post-training effectively. We observe that SFT models learn to predict the answer directly, avoiding step-by-step reasoning. As a result, during the RL post-training phase, the model does not generate the intermediate reasoning steps to solve the task. This reinforces the idea that passive pretraining (i.e., next-token prediction on data without explicit intermediate reasoning traces) overfits and constrains later exploration that prevents models from discovering more general reasoning patterns.

**Towards RL-Driven Pretraining** Building on these insights, we advocate for a new pretraining approach that emphasizes learning step-by-step reasoning patterns. Since we lack large-scale training data with explicit intermediate reasoning steps (e.g., Chain-of-Thought style demonstrations), we propose using RL to generate such reasoning traces through interaction with an environment and guided by appropriate reward functions. We posit that achieving truly general reasoning abilities requires training a model from scratch (i.e., RPT) to reason using reward supervision instead of the currently popular paradigm of supervised pretraining on passive data followed by finetuning for reasoning with rewards. However, this approach presents a significant exploration challenge because searching in the

space of language tokens from scratch is difficult. To realize this vision of RL-driven pretraining, we must efficiently explore and generate high-quality, coherent reasoning steps that are missing in current pretraining datasets. We provide ideas on addressing this challenge in Section 4.2.

## 4.2. Pretraining for Reasoning from Scratch (RPT) using Synthetic Tasks

> **Proposal**: We propose pretraining models on **synthetic tasks or games in constrained environments with reduced token spaces** to enable efficient exploration. Large token spaces hinder the discovery of reasoning patterns due to combinatorial complexity. By learning in simplified settings, models can acquire core reasoning skills that transfer to more complex domains like language.

| Model Configuration | Train (%) | Test (%) |
|---|---|---|
| Base Model | 88 | 89 |
| Base Model + SFT-then-RFT | **100** | 80 |
| Base Model + RFT | 94 | **93** |

*Table 3.* Comparing the effect of finetuning with RL (RFT) against supervised finetuning followed by RL finetuning (SFT-then-RFT) on the reasoning task of determining orthoganlity between vectors. We used Qwen 1.5B as the base model. Pure RFT training achieves the highest accuracy, while SFT models show signs of overfitting.

An important challenge when training a model from scratch with RL is efficiently exploring the space of reasoning traces to find correct solutions. The search space of natural language is nearly unconstrained, and therefore, the likelihood of finding a solution by generating a random combination of words is extremely small. The central question is: How can we make this search process more tractable?

To this end, we propose **training a "reasoning prior" on synthetic tasks** that incentivizes the learning of reasoning primitives within a smaller token space where exploration is much easier. By training on simplified tasks where the agent must iteratively interact with the environment to solve for a reward, the agent can develop key reasoning skills, such as hierarchical abstraction, causal reasoning, and compositional generalization, bypassing the complexities of natural language. Recent studies (Chan et al., 2022; 2024) suggest that capabilities such as ICL emerge from structural properties of natural language syntax (e.g., burstiness, where items cluster temporally or thematically, and Zipfian distributions) rather than specific semantics of language. This suggests that we can choose problems with drastically smaller token spaces that preserve key properties necessary for the emergence of reasoning prior. Once such reasoning prior is acquired, models can then be adapted to broader token

spaces and ultimately to language, integrating search and feedback mechanisms into large-scale language pretraining. The two open questions are: (i) Which tasks should be used for reasoning pretraining and (ii) How do we transfer reasoning prior to natural language? Some possibilities for (i) are generating logic games using a small set of randomly generated rules or using data sources with similar distributions to natural language but smaller token spaces. We can start with tasks with smaller token spaces and gradually increase the complexity with larger token spaces requiring more challenging reasoning. We see a parallel here between the state of self-supervised learning in computer vision a decade ago (Gui et al., 2024; Baradad et al., 2022; Walker et al., 2015; Jayaraman & Grauman, 2016; Agrawal et al., 2015; Doersch et al., 2015), where many attempts were made to develop a good proxy task for self-supervision. It took considerable research by the community to eventually find self-supervision objectives and tasks that eventually superseded the performance of supervised learning. We expect something similar here.

For (ii), we can draw to the common practice of adapting a neural network for a new computer vision task by preserving the early layers and potentially finetuning or re-initializing the last layers (Donahue et al., 2013; Agrawal et al., 2014). When it comes to reasoning prior, we expect it to be embedded in the intermediate layers of the neural network. To transfer the reasoning prior, we can preserve the intermediate layers but adapt the input/output layers to enable the reasoning to work in a new token space (Lu et al., 2021). Careful selection and design of the artificial environments (i) are essential to learning features and circuits that are useful and transferrable (ii) to natural language.

**Evidence from existing literature hinting at the plausibility of the proposed approach** We argue that the reasoning priors learned with synthetic tasks can plausibly transfer to natural language, supported by two key pieces of evidence. First, code is an example of an artificial environment that closely parallels language. Although code and language have different token distributions, pretraining on code before training on natural language has been widely used (Petty et al., 2024; Aryabumi et al., 2024; Kim et al., 2024). The structured nature of code – precise syntax, deterministic semantics, and modularity – enables models to develop strong priors for logic and problem-solving. Cognitive studies further support this approach: children who learn programming exhibit enhanced creativity, mathematical skills, and reasoning on other non-coding tasks (Scherer et al., 2019; Montuori et al., 2024). This suggests that structured environments, whether learned by humans or machines, facilitate the development of transferable reasoning skills.

Second, Zhong & Andreas (2024) demonstrated that even a random transformer, with only embedding layers optimized, can solve various algorithmic tasks, such as modular arith-

metic and parentheses balancing. This suggests that some meaningful circuits already exist within the architecture of random models. Based on this observation, we hypothesize that when a model is explicitly trained to learn to maximize rewards on synthetic tasks, it will develop more meaningful and transferrable circuits that can be transferred and used when interacting with the model in natural language.

Further, unlike real-world corpora, which conflate reasoning and knowledge – for example, in language, logical inference is often entangled with knowledge and contextual semantics – synthetic tasks enable explicit disentanglement of the two. In these tasks, knowledge may consist of predefined rules and primitives that govern system dynamics, and we have full control over them. Because these synthetic tasks remove the ambiguities and semantic dependencies inherent in language, models trained to solve such tasks focus solely on applying known rules to reason, solving for certain rewards by iteratively stepping through the environment. Thus, artificial environments serve as ideal testbeds for reasoning mechanisms that do not confound an agent's extent of a priori factual knowledge with its reasoning abilities.

### 4.3. Decoupling Knowledge and Reasoning to Generalize Across Domains

> **Proposal**: To enable robust generalization across knowledge domains, we propose **decoupling knowledge and reasoning** using an external memory bank to store knowledge and a **small context window**, allowing models to retrieve and process information selectively rather than the currently dominant paradigm of learning to reason using large context windows that can be more prone to overfitting.

The quest for disentangling knowledge and reasoning leads to the following questions: (i) What architectural priors foster more robust reasoning? (ii) How can we help the model generalize its reasoning to new problems and domains?

Current models discover reasoning by training a transformer model on long context. It is widely believed that larger models with longer context windows and more extensive data learn superior reasoning abilities as evidenced by their superior ICL (Dong et al., 2022) abilities. However, our experiments in Section 3 illustrate that simply scaling models with more data may not suffice to achieve general reasoning due to the intertwining of knowledge and reasoning. We hypothesize that training with longer context windows has higher chances of learning spurious correlations among past tokens to predict future tokens. This is evident in the *lost-in-the-middle* phenomenon, where models incorrectly ignore the content in the middle (Liu et al., 2024).

One way to encourage the emergence of more general reasoning is to train with *shorter context windows*. As the model needs to predict using fewer tokens, it will have lower chances to learn spurious correlations between tokens in the context window to predict future tokens. Crucially, human cognitive science studies provide an empirical grounding for this approach, suggesting that limiting context can foster more accurate and systematic reasoning (Miller, 1956; Newport, 1988; Elman, 1993; Cowan, 2001). In fact, as a result of the limited context, we develop strategies such as *chunking* (Thalmann et al., 2019), where we form abstractions that group related pieces of knowledge, enabling more complex concepts to be retained within working memory.

However, reducing the context window can prevent the model from having the long-range information necessary to make predictions. To mitigate this issue, we **propose to have an explicit *semantic memory* where data is stored and a learned retrieval mechanism to fetch a small amount of data** in the model's *working memory* (i.e., context window) on which the reasoning function operates. We include a high-level pseudocode illustrating our proposed hybrid memory + reasoning model in Appendix B.

We hypothesize that this combination of semantic memory for data storage, a learned data retrieval mechanism, and a small working memory on top of which reasoning operates will lead to a more scalable and generalizable reasoning system. The proposed architecture should allow for effective generalization to new problem domains by inserting the new knowledge in the semantic memory.

At a surface level, our proposal may seem akin to retrieval-augmented generation (RAG) (Lewis et al., 2021). However, RAG systems typically add the retrieved text into context and rely on ICL (Dong et al., 2022) over long sequences. This approach presents two key challenges: (i) it relies on a static, heuristic-based retrieval mechanism, such as nearest neighbors on a fixed embedding space, that does not allow the model to actively query, and (ii) it assumes the model has strong long-context reasoning abilities, which is often not the case. As context size grows, retrieval quality degrades, and models struggle to effectively utilize all retrieved information (Zhang et al., 2024).

Moreover, prior memory-based approaches, such as Guu et al. (2020) typically do not support the aforementioned dynamic, multi-round interactions with external memory. Yet, when tackling a challenging math problem, for instance, we often need to retrieve different relevant past problems or theorems at various stages of reasoning. As we progress, we may continue to refine our retrieval and also record the approach we use to solve the problem. This iterative retrieval and updating of knowledge is essential for more efficient learning.

In contrast, we propose to interact with the memory (read-/write) during *training* and *inference*, thereby learning to explicitly decouple memory from reasoning. Rather than appending all retrieved text into a long context window, we maintain a relatively *small* working memory to which the

model can repeatedly query and update an external semantic memory. Using a small context forces both the retrieval mechanism to retrieve more relevant information and for the reasoning module to operate on a smaller number of tokens and thereby have lesser chances of learning spurious correlations between tokens.

Conceptually, our proposed design is reminiscent of early neural memory architectures like Neural Turing Machines (Graves, 2014) and Differentiable Neural Computers (Graves et al., 2016), where a separate memory module is maintained. These approaches, however, often suffer from significant optimization instabilities (Aleš, 2016; Paaßen & Schulz, 2020) due to end-to-end differentiability across numerous memory-access steps, which often lead to volatile gradients. To circumvent such optimization bottlenecks, we propose *discrete* memory-access decisions trained via RL, thereby avoiding the vanishing gradients in long-horizon differentiable memory access. Also, since long-horizon RL optimization can be unstable, we propose leveraging a *curriculum* strategy to stabilize training by starting with simple memory interactions (read and write) and gradually progressing towards more complex, multi-step interactions.

## 5. Alternative Views

An alternative perspective to our proposal of disentangling knowledge and reasoning contends that knowledge and reasoning are so deeply intertwined that they are practically inseparable. For example, consider completing the sentence, "The coffee was unbearably hot, so he poured it into a ..." Although "metal cup" and "porcelain cup" may both be probable completions, most people would select "porcelain cup" based on the context that the coffee was hot. Yet, it is unclear whether such a choice arises from memorized associations (knowledge of common behavior) or inference (reasoning about likely scenarios). This fundamental ambiguity raises questions about whether knowledge and reasoning can or should even be disentangled in intelligent systems.

One perspective is that if an agent has to perform a task repeatedly, then it may not want to explicitly reason every time, but store the result of its reasoning as "knowledge". In such a scenario, knowledge and reasoning can be coupled for efficiency. Our argument is not that reasoning and knowledge cannot be coupled but that the agent should have a mechanism to separate them and not only reason based on correlations between its experiences so that it can truly apply the reasoning in a general way.

Furthermore, integrating RL into the pretraining of LLMs presents several significant challenges that question its feasibility and scalability. One major issue is the inefficiency of RL-driven training, which requires collecting data on the fly through interactions rather than leveraging static, large-scale corpora. This process is computationally expensive and risks becoming infeasible as model sizes and training demands grow. Finally, while we argue that supervised pretraining on passive data can limit exploration, training on synthetic tasks can suffer from a similar problem if they are poorly designed or constrained to narrow task distributions that ultimately do not transfer well to natural language. Particularly if real-world tasks in natural language include reasoning skills beyond the scope of synthetic pretraining tasks, then mechanisms to compensate for such differences in the pretraining and the needed capabilities will be required. Therefore, the choice of specific pretraining tasks is an important subject of future research.

## 6. Related Works

**Evaluating Reasoning in Large Language Models.** LLMs like GPT-4 (OpenAI, 2024), Llama 3 (Meta, 2024), and Qwen (DeepSeek-AI, 2024) show strong reasoning, but remain fragile, often overfitting to training patterns—as seen in our esoteric language experiments and corroborated by counterfactual and symbolic reasoning benchmarks (Mirzadeh et al., 2024; Wu et al., 2024). Fine-tuning has been shown to improve both memorization and generalization (Xie et al., 2024).

**Inference-time Scaling for Reasoning.** Inference-time methods include search-based approaches like CoT (Wei et al., 2023), ToT (Yao et al., 2024), and self-consistency (Wang et al., 2023a), which explore multiple reasoning paths. RL-based methods like RLHF (Ouyang et al., 2022; Han et al., 2024), ReST (Singh et al., 2024), and STaR (Zelikman et al., 2022; 2024) use search and reward signals (Lightman et al., 2023; Wang et al., 2024a; Uesato et al., 2022) to improve reasoning policies through iterative refinement.

**Memory Architectures.** While methods like RAG (Berges et al., 2024) provide static memory, and NTMs (Aleš, 2016) offer differentiable memory with training challenges, newer approaches embed memory in KV-caches (Berges et al., 2024) or separate reasoning and retrieval via special tokens (Jin et al., 2024). Yet, most methods still tightly couple reasoning with knowledge storage.

**Synthetic Data.** Synthetic data generation in language modeling typically relies on natural language prompts (Chen et al., 2024; HuggingFace, 2024; Microsoft, 2024). Prior work on symbolic synthetic tasks (Wu et al., 2022; 2021; Krishna et al., 2021) uses simple rule-based data, but often lacks scalability and complexity for emergent reasoning. Synthetic data has also shown promise in vision and RL domains (Wang et al., 2023b; Baradad et al., 2022; 2023; Wang et al., 2024b).

## Acknowledgment

We want to express our gratitude to Idan Shenfeld, Zhang-Wei Hong, Akarsh Kumar, and members of the Improbable AI lab for the helpful discussion on the paper. We are

grateful to MIT Supercloud and the Lincoln Laboratory Supercomputing Center for providing HPC resources. The research was supported in part by NSF CSGrad4US Fellowship, Google, and Amazon. The research was sponsored by the Army Research Office and was accomplished under Grant Number W911NF-21-1-0328. The research was also sponsored by the Office of Naval Research and was accomplished under Grant Number N00014-22-1-2740. This research was also sponsored by the United States Air Force Research Laboratory and the United States Air Force Artificial Intelligence Accelerator and was accomplished under Cooperative Agreement Number FA8750-19-2-1000. This research was supported also by the National Institute of Health under the Grant Number MH133066. The views and conclusions contained in this document are those of the authors and should not be interpreted as representing the official policies, either expressed or implied, of the Army Research Office, Naval Research Office, Air Force, or the U.S. Government.

## Author Contributions

**Seungwook Han** co-developed the project and contributed in all aspects of experiments and writing.

**Jyothish Pari** co-developed the project and contributed in all aspects of experiments and writing.

**Samuel Gershman** contributed to the paper's narrative and writing.

**Pulkit Agrawal** co-developed the project direction, advised SH and JP, and played a significant role in paper writing.

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

# A. Brainf**k and Befunge Experimental Details

## A.1. Example of Brainf**k Program

```
+++++++[>++++++++++<-]>+++++.
```

*Listing 3.* Brainf**k program to print the character 'K.'

## A.2. Example of Befunge Program

```
% &>:1-:v v *_$.@
%  ^    _$>\:^
%
```

*Listing 4.* Befunge program to calculate the factorial.

## A.3. Details on Experimental Protocol

The number of examples used for each language-task evaluation are as follows:

- Brainf**k Copy: 100

- Brainf**k Print: 676

- Brainf**k Sort: 100

- Befunge Print: 100

- Befunge Fibonacci: 1

- Befunge Factorial: 15

## A.4. Brainf**k Prompt

```
You are a helpful coding assistant.

Brainfuck is an esoteric programming language with a minimalist set of commands. It
    operates on an array of memory cells (initially all set to zero) and a data pointer
    that starts at the beginning of this array. The commands are as follows:

> - Move the data pointer to the right (increment the pointer to point to the next memory
    cell).
< - Move the data pointer to the left (decrement the pointer to point to the previous
    memory cell).
+ - Increment the byte at the data pointer (increase the value in the current memory cell
    by 1).
- - Decrement the byte at the data pointer (decrease the value in the current memory cell
    by 1).
. - Output the byte at the data pointer as an ASCII character (e.g., if the value is 65,
    it outputs 'A').
, - Read one byte of input and store its ASCII value in the byte at the data pointer.

Looping:
[ - If the byte at the data pointer is zero, jump forward to the command after the
    corresponding ']'.
] - If the byte at the data pointer is non-zero, jump back to the command after the
    corresponding '['.

Loops are used to iterate over a section of code until the byte at the data pointer
    becomes zero. They can be nested, but you must ensure each '[' has a matching ']'.

Memory and Data Pointer:
- Brainfuck operates on an array of memory cells (commonly 30,000 cells).
```

```
- All cells are initially set to zero.
- The data pointer can be moved left and right to access different cells.
- The '+' and '-' commands modify the value of the cell at the data pointer.
- You must manage memory manually, ensuring that the pointer does not move outside the
    bounds of the array.

ASCII Character Values:
- Lowercase letters have the following ASCII values:
  - 'a' = 97
  - 'b' = 98
  - 'c' = 99
  - 'd' = 100
  - 'e' = 101
  - 'f' = 102
  - 'g' = 103
  - 'h' = 104
  - 'i' = 105
  - 'j' = 106
  - 'k' = 107
  - 'l' = 108
  - 'm' = 109
  - 'n' = 110
  - 'o' = 111
  - 'p' = 112
  - 'q' = 113
  - 'r' = 114
  - 's' = 115
  - 't' = 116
  - 'u' = 117
  - 'v' = 118
  - 'w' = 119
  - 'x' = 120
  - 'y' = 121
  - 'z' = 122

### Instruction: Generate a Brainfuck program to print 'ey'

### Program: ++++++++++[>++++++++++<-]>+.<++++[>++++<-]>++++.

### Instruction: Generate a Brainfuck program to print 'ez'

### Program: ++++++++++[>++++++++++<-]>+.<++++[>++++<-]>+++++.

### Instruction: Generate a Brainfuck program to print 'hi'
```

*Listing 5.* Prompt for Brainf**k evaluation. This example is for the task of printing and the number of example instruction and programs pairs varies with the number of in-context example specified for evaluation. Only the instruction and program parts change for the other tasks of sorting and copying.

## A.5. Befunge Prompt

```
You are a helpful coding assistant.

Befunge is a two-dimensional, stack-based esoteric programming language where the
    instruction pointer (IP) moves across a grid in multiple directions. The commands
    control movement, stack manipulation, and program output. Here's a guide on how to use
     Befunge with examples:

1. **Grid and Execution Flow**:
Befunge code is written on a 2D grid. The IP starts at the top left and moves according to
    directional commands like '>' (right), '<' (left), '^' (up), and 'v' (down).

2. **Basic Commands**:
- **'+ - * /'**: Perform arithmetic with the top two stack values.
```

```
- **'>' '<' '^' 'v'**: Control the direction of the IP.
- **'_' '|'**: Conditional directions; '_' moves IP right if 0, left if non-zero; '|'
    moves IP down if 0, up if non-zero.
- **'@'**: Ends the program.
- **'"'**: Toggle string mode, pushing ASCII values of characters onto the stack.
- **'0-9'**: Push numbers onto the stack.
- **'.' ','**: Output values; '.' outputs a number, ',' outputs a character.
- **'#'**: Trampoline, which skips the next cell in the current direction.

3. **Stack Operations**:
- **':'** duplicates the top stack value, **'$'** removes the top stack value, and **'\'**
     swaps the top two values.

4. **Self-Modifying Code**:
Befunge allows the code to modify itself at runtime using the **'p'** (put) and **'g'** (
    get) commands, where 'p' writes to a grid position and 'g' reads from a position.

5. **Examples**:

- **Hello, World!**:
    ```
    >               v
    v",olleH">:#,_@
    >"dlroW",       ^
    ```

    Explanation: This program uses '"', ',', and '_' to output "Hello, World!" as it moves
        around the grid.

- **Addition of Two Numbers** (e.g., '3 + 5'):
    ```
    >3 5+ .@
    ```

    Explanation: '3' and '5' are pushed to the stack, '+' adds them, '.' outputs the
        result, and '@' ends the program.

- **Countdown from 9 to 0**:
    ```
    >9876543210v
    v:,_@       <
    ```

Explanation: Numbers 9 to 0 are pushed to the stack in reverse order. ':' duplicates the
    top value, ',' outputs it as a character, '_' checks if it's zero to change direction,
     and '@' terminates.

6. **Program Termination**:
Always use the '@' command to end execution.

### Instruction: Generate a Brainfuck program to print 'ey'

### Program: ++++++++++[>++++++++++<-]>+.<++++[>++++<-]>++++.

### Instruction: Generate a Brainfuck program to print 'ez'

### Program: ++++++++++[>++++++++++<-]>+.<++++[>++++<-]>+++++.

### Instruction: Generate a Brainfuck program to print 'hi'
```

*Listing 6.* Prompt for Brainf**k evaluation. This example is for the task of printing and the number of example instruction and programs pairs varies with the number of in-context example specified for evaluation. Only the instruction and program parts change for the other tasks of sorting and copying.

## A.6. Synthetic Task Details

We designed a synthetic math task to ensure that the base model, Qwen/Qwen2.5-1.5B-Instruct, could occasionally generate accurate reasoning traces while still producing incorrect ones. This was important for our RL training, as it ensured that the base model could explore correct reasoning traces with a non-trivial probability, making RL training significantly more efficient.

We created 100 training examples and 100 test examples. To evaluate the model's outputs, we developed custom verification code that processes the model's responses to determine whether they correctly answer the given question. Our verification system accepts multiple valid answer formats, ensuring flexibility in evaluation. Additionally, we manually post-filtered the evaluation outputs to correct cases where the verification process incorrectly classified a response.

We finetuned the Qwen/Qwen2.5-1.5B-Instruct model on 100 synthetic examples for 100 epochs using Low-Rank Adaptation (LoRA) (Hu et al., 2021) with rank=256, alpha=32, and a dropout rate of 0.05 applied to the query, key, and value matrices (Q, K, V). The training employed a cosine learning rate schedule with an initial learning rate of 5e-4, a batch size of 64, and 10 warmup steps. We used LoRA with these parameters for efficient training given our compute constraints. We recognize our SFT epochs are high for an LLM; however, we found that higher epochs were needed to reach decent test CoT accuracy.

For model training, we used LoRA with the following configuration:

*Listing 7.* LoRA Configuration for Qwen2-1.5B

```
lora_config = LoraConfig(
    r=256,
    lora_alpha=32,
    target_modules=["q_proj", "k_proj", "v_proj"],
    lora_dropout=0.05,
    bias="none",
    task_type="CAUSAL_LM"
)
```

The SFT training results are shown in Table 4:

|  | Train Acc CoT | Test Acc |
|---|---|---|
| SFT Epoch 100 | 100.0 | 82.0 |
| SFT Epoch 50 | 100.0 | 83.0 |
| SFT Epoch 20 | 74.0 | 62.0 |

*Table 4.* SFT Training Results

Subsequently, both the SFT and the base models underwent 31 steps of RL training using the RLOO algorithm (Kool et al., 2019). The RL hyperparameters included a batch size of 36, a single PPO epoch per iteration, and a KL coefficient of 0.5.

## A.7. Prompt Format and Sample Responses

*Listing 8.* Prompt and CoT sample response

```
[SYSTEM]
You are a helpful assistant. Think step by step concisely and answer the question in the
    following format: Solution: (...)
[USER]
Which of the vectors are orthogonal to [1, -1, -1]? (a) [3, 5, 0] (b) [-5, 2, 2] (c) [-1,
    -2, 4] (d) [1, 2, -1], output the answer as solution:
[ASSISTANT]
To determine which vectors are orthogonal to the vector \([1, -1, -1]\), we need to check
    if the dot product of each vector with \([1, -1, -1]\) is zero. Let's calculate the
    dot product for each vector:

1. For vector \([3, 5, 0]\):
    \[
    [1, -1, -1] \cdot [3, 5, 0] = 1 \cdot 3 + (-1) \cdot 5 + (-1) \cdot 0 = 3 - 5 + 0 = -2
        \neq 0
    \]
    So, \([3, 5, 0]\) is not orthogonal.
```

```
2. For vector \([-5, 2, 2]\):
   \[
   [1, -1, -1] \cdot [-5, 2, 2] = 1 \cdot (-5) + (-1) \cdot 2 + (-1) \cdot 2 = -5 - 2 - 2
      = -9 \neq 0
   \]
   So, \([-5, 2, 2]\) is not orthogonal.

3. For vector \([-1, -2, 4]\):
   \[
   [1, -1, -1] \cdot [-1, -2, 4] = 1 \cdot (-1) + (-1) \cdot (-2) + (-1) \cdot 4 = -1 + 2
      - 4 = -3 \neq 0
   \]
   So, \([-1, -2, 4]\) is not orthogonal.

4. For vector \([1, 2, -1]\):
   \[
   [1, -1, -1] \cdot [1, 2, -1] = 1 \cdot 1 + (-1) \cdot 2 + (-1) \cdot (-1) = 1 - 2 + 1 =
       0
   \]
   So, \([1, 2, -1]\) is orthogonal.

Therefore, the only vector that is orthogonal to \([1, -1, -1]\) is \([1, 2, -1]\).

Solution: d) [1, 2, -1]
```

*Listing 9.* Prompt and sample response from SFT model

```
[SYSTEM]
You are a helpful assistant. Think step by step concisely and answer the question in the
    following format: Solution: (...)
[USER]
Which of the vectors are orthogonal to [1, -1, -1]? (a) [3, 5, 0] (b) [-5, 2, 2] (c) [-1,
    -2, 4] (d) [1, 2, -1], output the answer as solution:
[ASSISTANT]
solution: (d)
```

## B. Pseudocode for Architecturally Decoupling Knowledge and Reasoning

### B.1. PT + RL Model Training Details

We train the PT + RL model in two stages.

**Stage 1: Supervised Pretraining (PT).** We collect 80,824 professional 9×9 Go game trajectories from online sources such as GoQuest (GoQuest, 2024) and other research archives (Müller, 2024; Brouwer, 2024) (averaging 47.6 moves each, including both players), giving roughly ∼1M training examples for next-move prediction. We train the network for 10 epochs with a batch size of 1024, a learning rate of $10^{-3}$, and weight decay of $10^{-4}$.

**Stage 2: Reinforcement Learning (RL).** We run self-play training à la AlphaZero. Each iteration generates 102,400 self-play games; for every move, we perform 32 MCTS simulations using Google DeepMind's `mctx` library (DeepMind, 2024). We use a batch size of 1024, a starting learning rate of $10^{-2}$ (with cosine decay over 200 total iterations), and weight decay of $10^{-4}$. This training procedure takes approximately 14 days on $4 \times$ A100 GPUs.

---

**Algorithm 1** Curriculum-Guided Reasoning with External Memory

---

 1: Initialize *ReasoningModel()*, *Memory()*, Curriculum ← [Easy, Med, Hard]
 2: mem ← Memory.reset() {external memory module}
 3: **for** stage in Curriculum **do**
 4:    **for** task in sample_tasks(stage) **do**
 5:       ctx ← [ ] {context for ReasoningModel}
 6:       **for** $t = 1$ to max_steps **do**
 7:          obs ← task at step $t$ {get current observation}
 8:          append obs to ctx {add current observation to context}
 9:          act ← ReasoningModel.query_action(ctx) {read, write, reason}
10:          **if** act.type = "READ" **then**
11:             append mem.read(act.key) to ctx {read from memory}
12:          **else**
13:             **if** act.type = "WRITE" **then**
14:                mem.write(act.key, act.value) {write to memory}
15:             **end if**
16:          **end if**
17:          outputs ← ReasoningModel.reason(ctx) {reason given current context}
18:          reward ← task.reward(outputs, mem.cost()) {compute reward}
19:          **if** task.done() **then**
20:             **break**
21:          **end if**
22:       **end for**
23:       ReasoningModel.update(query, act, reward) {reward-based learning objective}
24:    **end for**
25: **end for**

---

