# OpenReview forum: "Position: General Intelligence Requires Reward-based Pretraining"
_ICML.cc/2025/Position_Paper_Track — ICML 2025 Position Paper Track spotlightposter_

### Official Review · Reviewer_Qekj · 2025-03-06

**Significance:** 3
**Argument Clarity:** 4
**Rating:** 4
**Confidence:** 3

**Questions:**

See strengths and weaknesses

**Discussion Potential:**

2

**Paper Summary:**

This paper argues that in order to train LLMs with reasoning capabilities, we should pretrain models with RL from scratch rather than using RL after pretraining.  Additionally, the paper argues that in order to perform RL during pretraining, we should pretrain models in simplified environments with reduced token spaces in order to reduce the complexity of the search space.  Finally, the authors argue that we should decouple memory from reasoning by training separate modules for them in order to reduce overfitting to specific domains.

**Position:**

Yes

**Position In Title:**

Yes

**Related Work:**

3

**Strengths And Weaknesses:**

Strengths
- position is argued clearly
- The first argument of integrating RL into pretraining is well supported through experiments
- interesting position that can inspire discussion

Weaknesses
- towards the end of the position paper, the proposal and support becomes more vague.  For example, it is not clear to me what is meant by generating artificial environments.  Does it just mean that we generate data following specific properties rather than scraping all possible text off the internet for pretraining? In section 4.3 the authors mention some properties that a good memory architecture (write and read operations) should have and mention that reasoning would then be learning to interact with the memory, but it isn't clear how these write and read operations would fit in within the proposed direction of pretraining with RL objective.

**Support:**

2

---

> ### Author Rebuttal · Authors · 2025-03-31
>
> We thank the reviewer for their insightful comments and recognition of our paper's clarity and experimental support for integrating RL into pretraining. We appreciate the reviewer's acknowledgment of the novelty and discussion potential of our position on simplifying pretraining environments and decoupling memory from reasoning. Below, we address their comments in detail.
>
> > How might these artificial environments look like, and why would the skills transfer?
> >
> **Common Point #2 Raised by Reviewers 8iwJ, Qekj**
>
> We propose two potential directions for these artificial environments:
>
> 1. Constructing **symbol-based synthetic tasks** like LIME (Wu et al.) to teach core reasoning primitives, such as deduction, induction, and abduction, with simple hard-coded reward functions. Guo et al. have shown that even simple reward structures can guide a model towards better reasoning. Further work is required to extend prior works like LIME to enable scalable, automatic generation of different rules and reward functions.
> 2. **Partially-observed, interactive environments** (e.g., TextWorld, or LLM-generated tasks) designed to reward cognitive behaviors like verification, backtracking, subgoal-setting, and backward chaining (Gandhi et al.).
>
> Because **these tasks involve exploring, reasoning about hidden states, and forming strategic plans, models learn robust search and exploration strategies during pretraining**. Skills transfer naturally to real-world language tasks, where the same underlying reasoning and planning mechanisms apply. Prior studies (e.g., Cheng et al.) have shown some signs that finetuning on “Adversarial Taboo” game improves performance on standard benchmarks, and that enabling planning behaviors in negotiation tasks (Lewis et al.) likewise boosts effectiveness. By simultaneously training on such diverse environments altogether, we expect to cultivate generally applicable search and reasoning capabilities for downstream use.
>
> [1] Wu, Yuhuai, et al. "Lime: Learning inductive bias for primitives of mathematical reasoning." International Conference on Machine Learning. PMLR, 2021.
>
> [2] Guo, Daya, et al. "Deepseek-r1: Incentivizing reasoning capability in llms via reinforcement learning." 2025.
>
> [3] Gandhi, Kanishk, et al. "Cognitive behaviors that enable self-improving reasoners, or, four habits of highly effective stars." arXiv preprint arXiv:2503.01307 (2025).
>
> [4] Cheng, Pengyu, et al. "Self-playing adversarial language game enhances llm reasoning." *Advances in Neural Information Processing Systems* 37 (2024): 126515-126543.
>
> [5] Lewis, Mike, et al. "Deal or no deal? end-to-end learning for negotiation dialogues." arXiv preprint arXiv:1706.05125 (2017).
>
> > How does Section 4.3 on the hybrid memory+reasoning architecture fit within the proposed direction of RL pretraining?
> >
>
> The reviewer raises an important question about how the memory architecture (read/write operations) integrates with our proposed RL-based pretraining approach for reasoning.
>
> We posit that shorter context window training is needed for better reasoning training because it prevents models from picking up spurious correlations. However, this creates a fundamental problem: reasoning often requires synthesizing many pieces of information that can't all fit in a limited context window. Humans navigate this challenge through active memory management. Despite our limited working memory (Miller's law), we can effectively reason over larger information sets by replacing items in working memory with new ones retrieved from our larger memory bank.
>
> Similarly, for models to reason effectively while avoiding the pitfalls of overly large context windows, they need explicit read and write operations. These operations allow the model to actively edit the short-term context by accessing a larger memory repository. This ability to deliberate over which information to retrieve and store is central to reasoning itself.
> The **RL objective provides an ideal framework for training these memory operations because it can reward the model for developing effective strategies of memory interaction—** knowing when to write information, what to keep in active context, and when to retrieve previously stored knowledge. Previous works like Graves et al. 2014 [1] have explored other gradient-based methods, which have encountered significant optimization instability. To this end, we posit that the RL training paradigm may be well-suited for this problem.
>
> [1] Graves, Alex, Greg Wayne, and Ivo Danihelka. "Neural turing machines." arXiv preprint arXiv:1410.5401 (2014).

---

> > ### Comment · Reviewer_Qekj · 2025-04-02
> >
> > Thank you for the discussions, I believe my comments were addressed and will raise my score

---

### Official Review · Reviewer_xKuy · 2025-03-12

**Significance:** 4
**Argument Clarity:** 4
**Rating:** 5
**Confidence:** 4

**Questions:**

Training details for the RL experiments?
- Give more details to how the Go training is conducted. What data is pretrained on ? How is RL conducted? At least one paragraph would be nice
- Again, give more details for the SFT and RL experiments for the orthogonal experiments. what RL algorithm?

**Discussion Potential:**

4

**Paper Summary:**

This paper presents a position about LLM reasoning abilities, incorporating search into training. First, it shows that reasoning LLM's perform much better than non-reasoning LLM's on two programming tasks in esoteric languages that LLM's do not have a lot of training data for. It hypothesizes that integrating reasoning and search directly into pretraining will improve performance, and gives evidence for this hypothesis by comparing performance of an RL model, pretrained model, and pretrained then RL model on 9x9 Go. The RL-only model does best, while using a higher KL penalty with the pretrained + RL models decreases performance. Then it proposes potential methodologies for tractibly incorporating search/reasoning into pretraining of LLM's, including (1) using small artificial environments where task complexity can be managed and gradually increased in a curriculum, (2) decoupling knowledge and reasoning using a memory mechanism, using a language-based interface for managing memory in a working memory. It discusses alternative views at the end of the paper.

## Update after rebuttal
I maintain my enthusiastic review of this paper.

**Position:**

Yes

**Position In Title:**

Yes

**Related Work:**

3

**Strengths And Weaknesses:**

# Strengths
I really liked this paper! At first glance, the topic seemed like it might be cliche or obviously true, but the detailed argument of the paper gave a lot of insights and convincing arguments on the topic. I think it would be a great addition to the community's understanding of these issues.

The esoteric language puzzles is a great easy to understand benchmark that highlights how much better reasoning models are than non-reasoning models.

The 9x9 Go example and the orthogonality task do a great job giving evidence for the idea  that incorporating RL into the pretraining process, instead of tacking it on after pretraining, improves performance. Some further details here on the exact methodology (data used, model size, training tokens, RL algorithm, etc) would strengthen the evidence. Even better would be a link to code reproducing the experiments.

After the evidence provided, the paper has built up enough reputability to justify the speculation in the later paper on possible ways to incorporate reasoning and search into training. It's clear that there are technical challenges with these directions, but proposing them and giving evidence for their importance is a great use of a position paper!



# Weaknesses
There is a lack of detail for the experiments conducted. We don't need a full technical breakdown, but a few details would strengthen the convincingness of the evidence presented to make us believe in the results.

Some suggestions for improvement:
- In Table 1, report accuracy out of 100 scores as a fraction out of 1, such as 0.71 for 71% accuracy. this is easier to read and more in line with how people typically report accuracy.
- report average accuracy for reasoning vs nonreasoning models separately. saying avg accuracy is 12% seems misleading if o1 averages over 50% and all the other models get  < 13%.
- Give more details to how the Go training is conducted. What data is pretrained on ? How is RL conducted? At least one paragraph would be nice
- Again, give more details for the SFT and RL experiments for the orthogonal experiments. what RL algorithm?

**Support:**

4

---

> ### Author Rebuttal · Authors · 2025-03-31
>
> We thank the reviewer for their enthusiastic and detailed feedback. We appreciate their recognition that our work proposes “a great addition to the community’s understanding” and that our evidence “builds enough reputability to justify speculation on possible ways to incorporate reasoning and search into training.” We especially value their emphasis on the importance of proposing a **fundamentally different direction** for the community to engage with, since our position paper aims precisely to spark new thinking on how to embed reasoning, search, and dynamic memory into the core of model training.
>
> We address the reviewer’s specific questions below.
>
> > Improve Table 1 result presentation and separate out reasoning and non-reasoning models.
> >
>
> Thank you for these suggestions. We will revise the final camera ready version with the corresponding suggestions.
>
> > Give more details to how the Go training is conducted. What data is pretrained on ? How is RL conducted? At least one paragraph would be nice
> >
>
> We adopt a ResNet-based convolutional architecture similar to AlphaZero [1], using five residual blocks (ten convolutional layers) for the shared backbone between the policy and the value functions. The policy and value functions each have their own additional convolutional heads on top of this shared backbone.
>
> For the PT + RL model, we train this model in two stages:
>
> **Stage 1: Supervised Pretraining (PT).** We collect 80,824 professional 9×9 Go game trajectories collected from online from GoQuest [5] and different research groups [3,4]  (averaging 47.6 moves each, including both players), giving roughly ~1M training examples for next-move prediction. We train the network for 10 epochs with a batch size of 1024, a learning rate of 1e–3, and weight decay of 1e–4.
>
> **Stage 2: Reinforcement Learning (RL).** We run self-play training à la AlphaZero. Each iteration generates 102,400 self-play games; for every move, we perform 32 MCTS simulations. For MCTS, we use Google DeepMind’s mctx library  We use a batch size of 1024, a starting learning rate of 1e–2 (with cosine decay over 200 total iterations), and weight decay of 1e–4. This takes about 14 days on 4 × A100 GPUs.
>
> By contrast, the **Go 9×9 RL** baseline omits Stage 1 and directly uses the same RL hyperparameters as Stage 2. We will include these details in the final camera-ready manuscript.
>
> [1] Silver, David, et al. "Mastering chess and shogi by self-play with a general reinforcement learning algorithm." arXiv preprint arXiv:1712.01815 (2017).
>
> [2] https://github.com/google-deepmind/mctx
>
> [3] https://webdocs.cs.ualberta.ca/~mmueller/go/games.html
>
> [4] https://homepages.cwi.nl/~aeb/go/games/index.html
>
> [5] https://senseis.xmp.net/?GoQuest
>
> > Again, give more details for the SFT and RL experiments for the orthogonal experiments. what RL algorithm?
> >
>
> We finetuned the Qwen/Qwen2.5-1.5B-Instruct model on 100 synthetic examples for 100 epochs using Low-Rank Adaptation (LoRA) with rank=256, alpha=32, and a dropout rate of 0.5 applied to the query, key, and value matrices (Q, K, V). The training employed a cosine learning rate schedule with an initial learning rate of 5e-4, a batch size of 64, and 10 warmup steps.  We used LoRA [2] with these parameters to for efficient training given our compute. In addition, we recognize our SFT epochs are high for an LLM, however, we found that higher epochs were needed to reach decent test CoT accuracy.
>
> ```markdown
> |                | Train Acc CoT | Test Acc |
> |----------------|---------------|----------|
> | SFT Epoch 100  | 100.0         | 82.0     |
> | SFT Epoch 50   | 100.0         | 83.0     |
> | SFT Epoch 20   | 74.0          | 62.0     |
> ```
>
> Subsequently, both the SFT and the base models underwent 31 steps of RL training using the RLOO algorithm (Kool et al.). The RL hyperparameters included a batch size of 36, a single PPO epoch per iteration, and a KL coefficient of 0.5.
>
> We will release our code and checkpoints and data for camera ready.
>
> [1] Kool, Wouter, et al. "Buy 4 REINFORCE Samples, Get a Baseline for Free!" OpenReview, 2019, https://openreview.net/forum?id=r1lgTGL5DE.
>
> [2] Hu, Edward J., et al. "Lora: Low-rank adaptation of large language models." ICLR 1.2 (2022): 3.

---

### Official Review · Reviewer_T49Y · 2025-03-13

**Significance:** 3
**Argument Clarity:** 2
**Rating:** 3
**Confidence:** 4

**Questions:**

1. The experimental validation largely relies on synthetic tasks and esoteric programming languages. Could the authors elaborate on how these results are expected to generalize to more complex, real-world reasoning and decision-making tasks? What evidence do the authors have (or plan to collect) that supports the applicability of the authors' approach beyond controlled experimental settings?
2. Integrating RL directly into the pretraining phase is a bold proposal, but it raises concerns about computational feasibility at scale. How do the authors plan to address the potential exponential increase in computational cost, especially when training on massive, web-scale datasets typical for state-of-the-art LLMs?
3. The idea of decoupling memory from reasoning is central to the authors' position, yet the paper lacks sufficient architectural details. Could the authors provide a more concrete description—such as diagrams, pseudocode, or specific implementation strategies—that clarifies how the proposed hybrid memory-reasoning system will function and be stabilized during training?
4. While the analogy with AlphaGo versus AlphaZero is compelling, planning and reasoning in natural language domains differ substantially from board game strategies. How do the authors justify this analogy, and what specific aspects of LLM reasoning do the authors believe directly correlate with the exploration benefits observed in RL systems like AlphaZero?

**Discussion Potential:**

3

**Paper Summary:**

The paper argues that current large language models, despite impressive practical performance, are fundamentally limited in robust general reasoning because they **entangle memorized knowledge with the reasoning process**. It shows through experiments, e.g., esoteric programming languages and synthetic mathematical challenges, that models trained primarily via next-token prediction and subsequent RL are overfitted to their training data and struggle to transfer reasoning skills to new contexts. The authors propose a paradigm shift with three main arguments:
1. Integrating RL directly into the pretraining phase (rather than using supervised pretraining followed by RL), which they argue can encourage the exploration of novel reasoning strategies.
2. Using artificial environments with reduced token spaces to build transferable reasoning priors, thereby making search in language more efficient and disentangling reasoning from knowledge.
3. Developing hybrid architectures that decouple memory from reasoning, enabling dynamic interaction between the two via RL and curriculum learning, which should improve both long-horizon planning and adaptive knowledge integration.

## update after rebuttal

I'm standing with a positive view of this paper.

**Position:**

Yes

**Position In Title:**

Yes

**Related Work:**

3

**Strengths And Weaknesses:**

Strengths:
1. The paper unambiguously argues that the current training paradigm for LLMs is fundamentally limited because it “couples” knowledge and reasoning. It asserts that general reasoning can only emerge when search is integrated from the start—a position that is clearly stated in both the title and throughout the text.
2. The authors back up their claims with some experiments. For example, they use tasks in esoteric programming languages and synthetic mathematical reasoning challenges to demonstrate that standard next-token prediction leads to overfitting, while models enhanced with RL-based search show improved (though still limited) reasoning performance. This evidence is reminiscent of the contrast between AlphaGo and AlphaZero.
3. By challenging the dominant paradigm of supervised pretraining followed by RL fine-tuning, the paper is poised to generate constructive debate. Its discussion of alternative views and limitations invites further exploration and cross-examination by researchers in the field.

Weaknesses:
1. The proposal to integrate RL directly into the pretraining phase raises questions about computational cost and scalability. The paper could benefit from a more detailed discussion of how these approaches would scale to the massive datasets and model sizes typical in state-of-the-art LLM training.
2. The proposed hybrid architecture that decouples memory from reasoning is conceptually appealing but under-specified. More concrete implementation details, including algorithmic pseudocode or illustrative diagrams, would help clarify how dynamic memory interaction and RL-driven exploration would be achieved in practice.
3. While the analogy with AlphaGo versus AlphaZero is compelling, it may oversimplify the challenges inherent in language reasoning. The paper might further strengthen its position by addressing potential counterarguments that question the direct applicability of such analogies to the domain of natural language.

**Support:**

3

---

> ### Author Rebuttal · Authors · 2025-03-31
>
> We thank the reviewer for their thoughtful and detailed feedback. We appreciate their recognition that our paper “unambiguously argues that the current training paradigm for LLMs is fundamentally limited” due to the coupling of knowledge and reasoning, and that our esoteric programming experiments, reminiscent of AlphaGo versus AlphaZero, help illustrate the benefits of integrating search more directly. We address the reviewer’s questions below.
>
> > How to make RL pretraining scalable?
> >
>
> **Common Point #3 Raised by Reviewers 8iwJ, T49Y**
>
> We acknowledge that training with RL from scratch is challenging. Given the key ideas proposed in our paper, we believe it can still be made scalable and feasible.
>
> - Our AlphaGo vs. AlphaZero experiments show that AlphaZero can be trained with less compute and still beat AlphaGo (which relied on PT + RL).
> - From this, we also hypothesize that with RL training the models to **iteratively reason**, we **only require smaller models** and thus reduce overall compute requirements.
> - Further, our proposed hybrid memory-reasoning architecture enables more efficient RL training. **By shrinking the context window and training with an external memory**, O(N^2) complexity of transformers remains manageable, while memory lookups—O(N)—handle longer-term information.
> - Along with a staged curriculum that starts with simpler tasks and incrementally increases complexity, these architectural enhancements should together alleviate the costs of full-scale RL from the outset.
> - There are also recent works, such as Bai et al. 2024 [1] , that **parallelize and accelerate the runtime of RL environments, making RL pretraining an even more solvable** challenge.
>
> > How do the authors’ evaluation on estoeric programming languages and math generalize to other more complex, real-world tasks?
> >
>
> We refer the reviewer back to **Common Point #1 in the response to Reviewer 8iwJ.** In brief,  our esoteric programming language evaluations align with the findings of Wu et al. that showcase that LLMs’ reasoning fail to generalize across out-of-distribution scenarios (counterfactual tasks).
>
> > Details on training a hybrid memory + reasoning model
> >
>
> We are including high-level psuedo-code illustrating the hybrid memory + reasoning model. . The system follows a curriculum of tasks that gradually increases the complexity of memory interactions. The reasoner receives observations, issues read/write actions on the external memory, and then reasons over the combined context. A reward function factoring in task performance and memory overhead guides learning of the read/write policy, fostering a clear separation between memory and reasoning.
>
> ```python
> init ReasoningModel(), Memory(), Curriculum = [Easy, Med, Hard]
>
> mem = Memory.reset() # external emory module
> for stage in Curriculum:
>     for task in sample_tasks(stage):
>         ctx = [] # context for ReasoningModel
>
>         for t in range(max_steps):
>             obs = task[t] # get current observation
>             ctx.append(obs) # add current observation to conte
>             act = ReasoningModel.query_action(ctx) # read, write, reason
>
>             if act.type == "READ": # read from memory
>                 ctx.append(mem.read(act.key))
>             elif act.type == "WRITE": # write to memory
>                 mem.write(act.key, act.value)
>
> 	          outputs = ReasoningModel.reason(ctx) # reason given current context
> 	          reward = task.reward(outputs, mem.cost()) # compute reward
> 	          if task.done():
> 	            break
>
>         ReasoningModel.update(query, act, reward) # reward-based learning objective
> ```
>
> > Though the analogy to AlphaZero is compelling, could the authors clarify which aspects of LLM reasoning align with the exploration advantages seen in systems like AlphaZero?
> >
>
> We propose training models in interactive, partially observed environments (e.g., TextWorld or LLM-generated tasks) designed to **reward cognitive behaviors—verification, backtracking, subgoal setting, and backward chaining** (Gandhi et al., 2025). These tasks encourage models to **develop robust exploration and planning strategies that generalize to real-world reasoning.** Prior studies support this approach: Cheng et al. (2024) found fine-tuning on adversarial language games improved benchmark performance, and Lewis et al. (2017) demonstrated strategic planning enhanced negotiation effectiveness. Training simultaneously on diverse tasks should foster broadly applicable reasoning capabilities.
>
> **We refer the reviewer to Common Point #2 addressed in the response to Reviewer Qekj for a more detailed discussion.**
>
> [1] Gandhi, Kanishk, et al. "Cognitive behaviors that enable self-improving reasoners, or, four habits of highly effective stars." 2025.
>
> [2] Cheng, Pengyu, et al. "Self-playing adversarial language game enhances llm reasoning." NeurIPS 2024.
>
> [3] Lewis, Mike, et al. "Deal or no deal? end-to-end learning for negotiation dialogues." 2017.

---

### Official Review · Reviewer_8iwJ · 2025-03-14

**Significance:** 3
**Argument Clarity:** 3
**Rating:** 4
**Confidence:** 4

**Questions:**

How do we train with RL from the start? What kind of reward signals would guide an LLM without human-labeled data?
What do these artificial environments look like? Are they structured like programming tasks, games, or something else? How do we ensure skills transfer to real-world language tasks?
How do we prevent new reasoning traps? Could models still get stuck in loops, just in different ways?
What’s the computational cost? RL is expensive—how can we make this training method scalable?
How do we evaluate success? If reasoning is learned through search, what benchmarks prove it’s actually improving?
What about safety? How do we ensure models exploring freely don’t generate harmful or biased outputs?

**Discussion Potential:**

4

**Paper Summary:**

This paper argues that LLMs struggle with real reasoning because they rely too much on memorized patterns. Testing them on esoteric programming languages reveals they can’t apply their knowledge flexibly, proving their reasoning is too tied to training data.
The authors claim the current training approach, supervised pretraining + RL fine-tuning, is limiting. Like AlphaZero outperformed AlphaGo by learning through self-play, LLMs need search and exploration from the start, not just as an afterthought. They propose three fixes: (1) Use RL during pretraining to encourage exploration, (2) Train in simplified artificial environments to develop reasoning skills, and (3) Decouple memory from reasoning so models can dynamically store and retrieve knowledge.
Bottom line: If we want AI to truly reason, we need to train it differently, by making reasoning an active, exploratory process from day one.

**Position:**

Yes

**Position In Title:**

Yes

**Related Work:**

3

**Strengths And Weaknesses:**

Strengths
This paper takes a clear stance and presents a well-structured argument on why LLMs struggle with reasoning. The esoteric programming test is a clever way to expose their limitations, and the AlphaZero analogy makes the case for search-based training easy to grasp. The authors don’t just critique current methods, they propose three concrete solutions (RL pretraining, artificial environments, memory decoupling), giving clear directions for future research. The paper is well-written, engaging, and timely, making it a strong discussion starter for advancing AI reasoning.
Weaknesses
The proposed solutions, while intriguing, are mostly theoretical, there’s no empirical proof that RL pretraining or artificial environments will fix the issue. The esoteric language test is narrow in scope, so it’s unclear if the problem applies broadly across all reasoning tasks. Also, search-based training at scale is expensive, and the paper doesn’t fully address how to make it efficient. While the ideas are exciting, the practical challenges of implementation remain open-ended.

**Support:**

3

---

> ### Author Rebuttal · Authors · 2025-03-31
>
> We thank the reviewer for their detailed assessment. The reviewer says that our evaluation on esoteric programming languages is “a clever way to expose their limitations. and that the AlphaZero analogy “makes the case for search-based training easy to grasp.”
>
> > The esoteric language test is narrow in scope, so it’s unclear if the problem applies broadly across all reasoning tasks.
> >
>
> **Common Point #1 Raised by Reviewers 8iwJ, T49Y**
>
> Our esoteric programming language test aims to evaluate a model’s capacity to handle rarely or unseen tasks, highlighting lack of generalization of current LLMs reasoning. These results **align with many findings from Wu et al.** 2024 [1], which reveal that LLMs  can struggle with reasoning in a variety of counterfactual tasks (such as Python with 1-indexing). This points to a more **general challenge in the reliability of reasoning mechanisms across out-of-distribution scenarios**.
>
> [1] Wu, Zhaofeng, et al. "Reasoning or reciting? exploring the capabilities and limitations of language models through counterfactual tasks." NAACL. 2024.
>
> > The proposed solutions, while intriguing, are mostly theoretical, there’s no empirical proof that RL pretraining or artificial environments will fix the issue.
> >
>
> Preliminary evidence from AlphaGo vs. AlphaZero experiments (Section 4.1, Figure 2) and synthetic math tests with pretrained LLMs (Section 4.1, Table 3) suggests RL-only approaches (AlphaZero) can outperform PT+RL methods (AlphaGo) with less compute, as shown in Silver et al.'s Figure 3. While these simple experiments perform some validation, we agree that further investigation is required to more strongly validate this hypothesis.
>
> [1] Silver et al*.* ”Mastering the game of Go without human knowledge.” Nature **2017.
>
> > Search-based training at scale is expensive and practical implementation remains open-ended.
> >
>
> We understand the reviewer’s concern and refer them to **Common Point #3 in the response to Reviewer T49Y**, where we outline how to keep search-based training feasible at scale—by employing smaller context windows, external memory, staged curricula, and parallel environment simulations. These strategies aim to mitigate the computational burden of RL-based training and make it a more practical approach in real-world settings.
>
> > What do these artificial environments look like?
> >
>
> We refer to the **Common Point #2 addressed in the response to Reviewer Qekj f**or a detailed discussion. Briefly, we propose **partially observed, interactive, and simplified text-based environments** (e.g., TextWorld) that inherently **require cognitive behaviors—verification, backtracking, subgoal creation, and backward chaining**—to help models learn robust search and reasoning methods transferable to real-world tasks. Additionally, we propose s**ymbolic synthetic tasks with simple hard-coded rewards** to teach core reasoning primitives like deduction, induction, and abduction.
>
> > How do we evaluate success?
> >
>
> To gauge whether search-based training truly improves reasoning, one can employ *established benchmarks*—for instance, math (e.g., HARDMath [1] ~60% accuracy), coding (e.g., SWEBench [2] 65% accuracy), and real-world agentic tasks (e.g., AitW General and Web Shopping [3] 50% accuracy)—to assess *accuracy*, *robustness*, and *adaptability* against standard baselines.
>
> **Additionally, out-of-distribution (OOD) generalization is a crucial measure of reasoning progress**, requiring models to handle tasks unlike their training data. We need *new benchmarks* targeting OOD reasoning—like *esoteric programming languages, counterfactual tasks, or data-limited physics domains* (e.g., photonic crystals)—to test models' adaptation beyond familiar patterns.
>
> [1] Cobbe, Karl, et al. "Training verifiers to solve math word problems." 2021.
>
> [2] Jimenez, Carlos E., et al. "Swe-bench: Can language models resolve real-world github issues?." 2023.
>
> [3] Rawles, Christopher, et al. "Androidinthewild: A large-scale dataset for android device control." NeurIPS 2023.
>
> > What about safety?
> >
>
> We agree safety is paramount. Many biases arise from coupling knowledge and reasoning—for instance, models trained on large text corpora often internalize stereotypes (“doctors are typically male”). **Our approach decouples these processes by training on structured tasks or games, limiting exposure to stereotypes**. Nonetheless, real-world data remain essential, and thereby our method along with existing alignment methods (e.g., bias mitigation, red-teaming, oversight) [1,2,3,4] can result in safer and more aligned outputs.
>
> [1] Hong, Zhang-Wei, et al. "Curiosity-driven red-teaming for large language models." ICLR 2024.
>
> [2] Zhou, Chunting, et al. "Lima: Less is more for alignment." NeurIPS 2023.
>
> [3] Lin, Stephanie, Jacob Hilton, and Owain Evans. "Truthfulqa: Measuring how models mimic human falsehoods." ACL 2021.
>
> [4] Bai, Yuntao, et al. "Constitutional ai: Harmlessness from ai feedback." 2022.

---

### Decision · Program_Chairs · 2025-04-29

**Decision:**

Accept (spotlight poster)

**Comment:**

Overall, the position presented in this paper is timely, in that reasoning LLMs have come to the forefront and are "frontier" models for many tasks, especially tasks related to advanced coding and advanced mathematics. The position in the paper is well stated and supported, and very likely to stimulate additional discussion among the broader community.

As one reviewer pointed out, even though this is a timely topic and there is much discussion in online forums, the paper presents several insights, (e.g., through the experiments with esoteric programming languages, or Go) which go well beyond the broader community discourse and support the position laid out in this paper.

The paper could be strengthened by directly addressing alternate views in Section 5.